# Development of mental health first aid guidelines for psychosis for Brazil: A Delphi expert consensus study

Simone Scotti Requena[1], Alexandre Andrade Loch[2,3], Kathlen Nataly Mendes[2], Nicola J. Reavley[1] *

1 Centre for Mental Health and Community Wellbeing, Melbourne School of Population and Global Health, The University of Melbourne, Melbourne, Victoria, Australia, 2 Laboratorio de Neurociencias, Instituto de Psiquiatria, Hospital das Clinicas da Faculdade de Medicina, Universidade de Sao Paulo, Sao Paulo, Sao Paulo, Brazil, 3 Instituto Nacional de Biomarcadores em Neuropsiquiatria, Conselho Nacional de Desenvolvimento Cientifico e Tecnologico, Sao Paulo, Sao Paulo, Brazil

* nreavley@unimelb.edu.au

**Data Availability Statement:** The data supporting our findings is attached as the Supporting Information file 1, which contains all the

## Abstract

Psychotic symptoms can be highly debilitating for those experiencing them. Community members, including family and friends, can play a crucial role in providing support to a person during the early stages of psychosis, provided they have the necessary resources. Mental health first aid guidelines for psychosis have been developed for high-income countries and this study aimed to adapt those guidelines for Brazil. A Delphi expert consensus method was used to gather the views and opinions of 28 health professionals and 24 individuals with lived experience of psychosis in Brazil over two survey rounds. Firstly, 403 statements were translated from English to Brazilian-Portuguese. In the Round 1 survey, participants were asked to rate each statement based on how important they believed it was for it to be included in the Brazilian guidelines. They were also asked to suggest new actions if they wished. Consensus was reached on 257 statements. Eight new statements were created and endorsed from panelists' comments, and a further 45 statements were unique to the Brazilian guidelines. There was a modest level of similarity between the English-language and Brazilian guidelines. However, the Brazilian guidelines had a greater focus on the importance of family support for people with psychosis and stigma as a possible barrier for openly discussing help-seeking actions for mental health problems in Brazil.

## Introduction

In 2019, the global incidence rate of individuals experiencing psychosis was estimated to be 26.6 per 100,000 person-years, with significant geographical variation [1]. In São Paulo, the largest Brazilian city, the estimated incidence of first-episode psychosis was 15.8 per 100,000 person-years in 2007 [2], whereas in other municipalities, this was 21.3 per 100,000 person-years more recently [3]. People with psychosis tend to have poorer physical health, feel more isolated and disconnected, have lower rates of employment [4], and face an increased risk of

statements that were presented to panelists, including their ratings.

**Funding:** Mental Health First Aid International. The funders had no role in study design, data collection and analysis, decision to publish, or preparation of the manuscript.

**Competing interests:** The authors have declared that no competing interests exist.

suicide [5], all of which can be highly debilitating for both those affected and those close to them. Acute psychosis carries the highest disability weight among all mental health problems [5]. Therefore, timely support is of extreme importance.

It is estimated that less than one-third of the global population living with psychosis receive mental health treatment, with far lower coverage in low-and-middle-income countries (LMICs) than in high-income countries (HICs) [6]. One of the major factors contributing to low mental health treatment rates in LMICs is the lack of services or resources. However, poor levels of mental health literacy and high levels of stigma towards people with mental health problems also play crucial roles in the low rates of mental health treatment in LMICs [7–10].

Various studies conducted in Brazil have explored stigma and discrimination towards individuals with schizophrenia, commonly associated with psychosis. One study involving 500 participants found that half of them perceived individuals with schizophrenia as capable of provoking negative emotions, while nearly two-thirds viewed them as dangerous [11]. Another study, which included nearly 2,500 participants from the community as well as psychiatrists, revealed that stigma towards people with schizophrenia increased with the ability of community members to recognize the condition, whereas psychiatrists showed the highest scores in negative stereotyping and perceived prejudice towards people with schizophrenia [12]. Another study investigated potential risk factors associated with hospital readmission in individuals with bipolar and psychotic disorders [13]. This study found that family's agreement with continuing hospitalization predicted hospital readmission, with families of readmitted individuals perceiving them as dangerous and unhealthy. These studies provide evidence of the discrimination individuals with psychosis experience in Brazil. Therefore, improving mental health literacy may be a crucial step forward.

One way to improve knowledge about psychosis and reduce stigma in the Brazilian community is to enable individuals to recognize when someone is experiencing psychosis, so they can encourage early help-seeking. This is the aim of the Mental Health First Aid (MHFA) training program, which was created in Australia in 2000 to teach individuals how to assist someone who is developing a mental health problem or is in a mental health crisis until appropriate professional help is received or the crisis ends [14].

MHFA training is based on Delphi expert consensus studies involving mental health professionals and people with lived experience [15]. A 2018 systematic review (18 samples; 5,936 participants) found that the MHFA training effectively improved mental health literacy, reduced stigma, and promoted help-seeking among individuals with mental health problems [15]. However, most of these studies were conducted in HICs and may not be generalizable to LMICs like Brazil, which has different cultures and health systems [16]. For instance, the cultural adaptation of the mental health first aid guidelines for psychosis in China found that family involvement was rated as being more important than in the guidelines for English-speaking countries [17]. Similarly, the cultural adaptation of the mental health first aid guidelines for suicide in Brazil highlighted the importance of family and friends when assisting a person at risk of suicide [18]. Therefore, this study aimed to develop mental health first aid guidelines for psychosis in Brazil using the Delphi expert consensus methodology.

## Materials and methods

This study used the Delphi expert consensus method to develop the Brazilian mental health first aid guidelines for psychosis. This method involves gathering expert opinions and assessing the consensus among groups of individuals with diverse experience in a given field [19]. Although a variety of expertise is not mandatory for the Delphi method, it is crucial to consider diversity when selecting panel members to ensure optimal decision-making, including

individuals with lived experience and a diverse range of health professionals [19]. This study was conducted in four stages: (1) survey development, (2) identification and recruitment of expert panels, (3) data collection and analysis, and (4) guidelines development.

## Survey development

A senior psychiatrist (AAL) translated the English-language Round 1 survey of the mental health first aid for psychosis study [20] to the Brazilian-Portuguese language (Note: the Round 1 survey was used as the English-language guidelines were in the process of re-development and the final version was not available). Whenever necessary, minor edits were made to statements to improve language clarity and relevancy to the Brazilian culture; for instance, the term "nuts" from the English-language statement "The first aider should try to avoid using stigmatizing terms that may make the person feel defensive, e.g. crazy, nuts, psycho" was deleted as this term is not used in the same context in the Brazilian-Portuguese language. Also, the English-language statement "If the person is having difficulty getting advice or help, the first aider should encourage the person to contact a mental health advocacy or support agency" was reworded as "If the person is having difficulty getting advice or help, the first aider should encourage the person to contact a mental health advocacy or support agency, such as a Center for Psychosocial Care", known as "CAPS", which are mental health support services in Brazil.

Furthermore, all English-language statements under the headings "What to do if the person is in a severe psychotic state" and "What to do if the person behaves aggressively" (136 statements) were combined into one heading as "What to do if the person is in a severe psychotic state or behaving aggressively" (68 statements) in the Brazilian survey to shorten this.

A total of 403 statements were added to the Brazilian Round 1 survey (see S1 File). Participants were instructed to rate each statement according to how important they believed it was to be included in the mental health first aid guidelines for psychosis in Brazil. The statements were organized into the following sections: Recognizing and acknowledging that someone may be experiencing psychosis; Approaching the person; Communication (non-crisis situation); Talking with the person (non-crisis situation); Communication difficulties; Being supportive; Substance use; Postnatal psychosis; Encouraging professional help (non-crisis situation); If the person doesn't want professional help (non-crisis situation); Hallucinations and delusions (non-crisis situation); Assessing whether the person is in crisis; When the person is in crisis (severe psychotic state or behaving aggressively); Severe psychotic states (crisis situation); Aggression (crisis situation); Self-care for the first aider. After each section, there was one open-ended text box with the following question "Would you like to suggest another action?".

## Identification and recruitment of expert panels

To ensure a diversity, this study had two expert panels: (1) Professional; (2) Lived experience, following the procedures used in Bond et al.'s study conducted in English-speaking countries [20]. To be eligible to participate, individuals had to be 18 years old or over, and:

• Professional panel: To be a health care practitioner with experience in psychosis.

• Lived experience panel:

 a. To have a lived experience of psychosis, or

 b. To be caring for someone experiencing psychosis.

There were no exclusion criteria regarding the level of experience within panels, to ensure an appropriate level of diversity within each panel. Potential participants were identified and invited by phone, email, or face-to-face to take part in the study. For the professional panel,

individuals from universities, including clinicians and researchers, and hospitals specialized in treating people experiencing psychosis (i.e., psychiatry departments of USP and UNIFESP; public universities/hospitals in Brazil) were contacted. For the lived experience panel, outpatients from specialized clinics from USP and UNICAMP, social media support groups, and specialized non-governmental organizations were contacted. To increase the number of participants, confirmed and potential participants were asked to inform others about the study. Potential participants were given a brief description of the study, either verbally or by email, and then sent the survey link.

### Data collection and analysis

Data were collected over two survey rounds. Eligible participants received the survey link hosted by online survey software. The first pages of the Round 1 survey included a detailed description about the study, questions confirming eligibility for participation, and consent for participation as a tick box. The following pages included socio-demographic questions such as age, gender, field and setting of practice (if applicable), and primary experience with psychosis (professional, lived, or caretaker).

All statements were rated on a five-point Likert scale: (1) Essential, (2) Important, (3) Depends/Don't know, (4) Unimportant, (5) Should not be included. Statements rated as "Essential" or "Important" by 80% or above of participants in each panel were directly included in the final guidelines as they reached 80% consensus between panels. If statements were rated as "Essential" or "Important" by 70–79% of participants in each panel, those statements were re-rated at the Round 2 survey. Statements rated as "Essential" or "Important" by less than 70% of participants from either panel were immediately excluded from the final guidelines. The cut-off values were based on the English-language redeveloped mental health first aid for psychosis Delphi study [20] as well as the culturally adapted mental health first aid guidelines for depression [21] and problem drinking [22] in Brazil.

Proposed new statements from participants' Round 1 open-ended responses were created by AAL and added to the Round 2 survey. We note that 46 statements were missed from the Round 1 survey due to a technical issue, and 18 statements were missed from the Round 2 survey due to the inclusion of duplicate values in the data set; these were rectified, and an additional survey was sent to participants (see S1 File).

### Guidelines development

Statements that reached consensus at Rounds 1 and 2, i.e., those rated as "Essential" or "Important" by ≥80% participants in both panels, were assembled into sections (see S1 File). Whenever necessary, SSR and NR edited proposed new statements to improve clarity. Two authors who were native speakers of the Brazilian-Portuguese language (KNM, SSR) checked the final list of included statements. Then, the final statements were assembled into a final document (see S2 File).

### Ethics statement

This research was approved by the University of Melbourne Human Research Ethics Committee (HREC No.1852131.1) and by the University of Sao Paulo Ethics Review Committee (Approval No. 20017419.5.0000.0068). All research was conducted in accordance with the National Health and Medical Research Council's National Statement on Ethical Conduct in Human Research and the Declaration of Helsinki. Participants were given a detailed plain language statement prior to participation, and written informed consent for participation was obtained from all participants as a tick box at the start of the survey.

## Results

### Expert panel information

Fifty-two participants (28 professionals, 24 people with lived experience) completed the Round 1 survey, and 24 participants (14 professionals, 10 people with lived experience) completed the Round 2 survey (see Table 1 for retention rates).

Participant socio-demographic characteristics by panel are shown in Table 2. In the professional panel, most of the participants were women (75%) aged 19–62 (mean ± SD = 36.7 ± 11.5). Similarly, in the lived experience panel, most participants were women (88%) aged 23–64 (mean ± SD = 42.1 ± 12.7). In the professional panel, one-third of participants reported their primary expertise with psychosis as psychologists. Half of the professionals said their primary area of practice involved working in a psychiatric hospital or psychiatry department within a hospital. In the lived experience panel, one-third of participants said their primary experience with psychosis was caring for someone with psychosis.

### Rating of statements

Expert panels agreed on 257 statements for inclusion in the Brazilian mental health first aid guidelines for psychosis over two survey rounds (see S1 File). Fig 1 shows the number of statements endorsed, excluded, and re-rated at both rounds.

Four-hundred and three statements were rated in the Round 1 survey, with 221 endorsed, 66 re-rated, and 116 excluded (see S1 Table in S1 File). The Round 2 survey included 60 new statements created from participants' comments and 66 statements that needed to be re-rated from round 1. Of these 126 statements at the Round 2 survey, 36 were endorsed and 90 were excluded (S1 File). A total of 257 statements (i.e., 249 original and 8 new statements) were included in the Brazilian guidelines for psychosis (see S1 File).

Expert panels had similar average ratings at both rounds (80–81% at round 1 and 75% at round 2). The Spearman's correlation coefficient between the two panels' ratings was 0.73 at round 1 ($P < 0.001$). The correlation coefficient for round 2 ratings was not calculated due to the uneven drop-out rates between panels.

### Differences between the English-language and Brazilian guidelines

A total of 96 out of 403 (24%) statements had different ratings in the Brazilian and the final English-language guidelines (see S1 File for a comparison between these guidelines). Of the 96 statements, 51 (53%) of them were excluded from the Brazilian guidelines but endorsed in the English-language guidelines. Additionally, eight new statements were added to the Brazilian guidelines across various sections:

- **Approaching the person** (n = 3):

    1. The first aider should pay attention to non-verbal communication signs from the person.

    2. The first aider should not be physically or verbally aggressive with the person.

**Table 1. Participant retention rates across two survey rounds.**

| Expert panel | Round 1 | Round 2 | Retention rate (%) |
|---|---|---|---|
| Professional | 28 | 14 | 50 |
| Lived experience | 24 | 10 | 42 |
| Total | 52 | 24 | 46 |

**Table 2. The socio-demographic characteristics of participants.**

| | Professional panel | | Lived experience panel | |
|---|---|---|---|---|
| | Frequency | Percentage | Frequency | Percentage |
| | (n = 28) | (%) | (n = 24) | (%) |
| **Gender** | | | | |
| Woman | 21 | 75.0 | 21 | 87.5 |
| Men | 7 | 25.0 | 3 | 12.5 |
| **Age (years)** | | | | |
| Min-max | 19 | 62 | 23 | 64 |
| Mean ± SD | 36.7 | 11.5 | 42.1 | 12.7 |
| **Caretaker** | - | - | 8 | 33.3 |
| **Primary expertise** | | | | |
| Psychologist[a] | 9 | 32.1 | NA | NA |
| Psychiatrist[b] | 4 | 14.3 | NA | NA |
| Missing | 4 | 14.3 | NA | NA |
| Researcher[c] | 3 | 10.7 | NA | NA |
| Nurse[d] | 3 | 10.7 | NA | NA |
| Social worker | 3 | 10.7 | NA | NA |
| Other[e] | 2 | 7.1 | NA | NA |
| **Setting of practice** | | | | |
| Psychiatry department[f] | 7 | 25.0 | NA | NA |
| Psychiatric hospital | 7 | 25.0 | NA | NA |
| CAPS[g] | 4 | 14.3 | NA | NA |
| Other health departments[h] | 4 | 14.3 | NA | NA |
| Private practice | 3 | 10.7 | NA | NA |
| Department of health[i] | 2 | 7.1 | NA | NA |
| Missing | 1 | 3.6 | NA | NA |

Data collected in Round 1.

[a]Included one intern.

[b]Included two interns.

[c]Included one assistant.

[d]Included two technicians.

[e]Included one occupational therapist and one health worker assistant.

[f]Department within a hospital-university.

[g]CAPS are "Centers for Psychosocial Care" part of the Brazilian public health system.

[h]Other health departments within an educational institution.

[i]Government Department of Health.

3. The first aider should be as friendly as possible.

- **Communication (non-crisis situation)** (n = 1):

4. The first aider should not discuss any beliefs about religion and mental health with the person.

- **If the person doesn't want professional help (in a non-crisis situation)** (n = 1):

5. The first aider should keep a list of emergency numbers, such as the Unidade de Pronto Atendimento (UPA 24h), as well as psychiatric emergency numbers (note. "UPA 24h" is part of the public Emergency Care Network in Brazil).

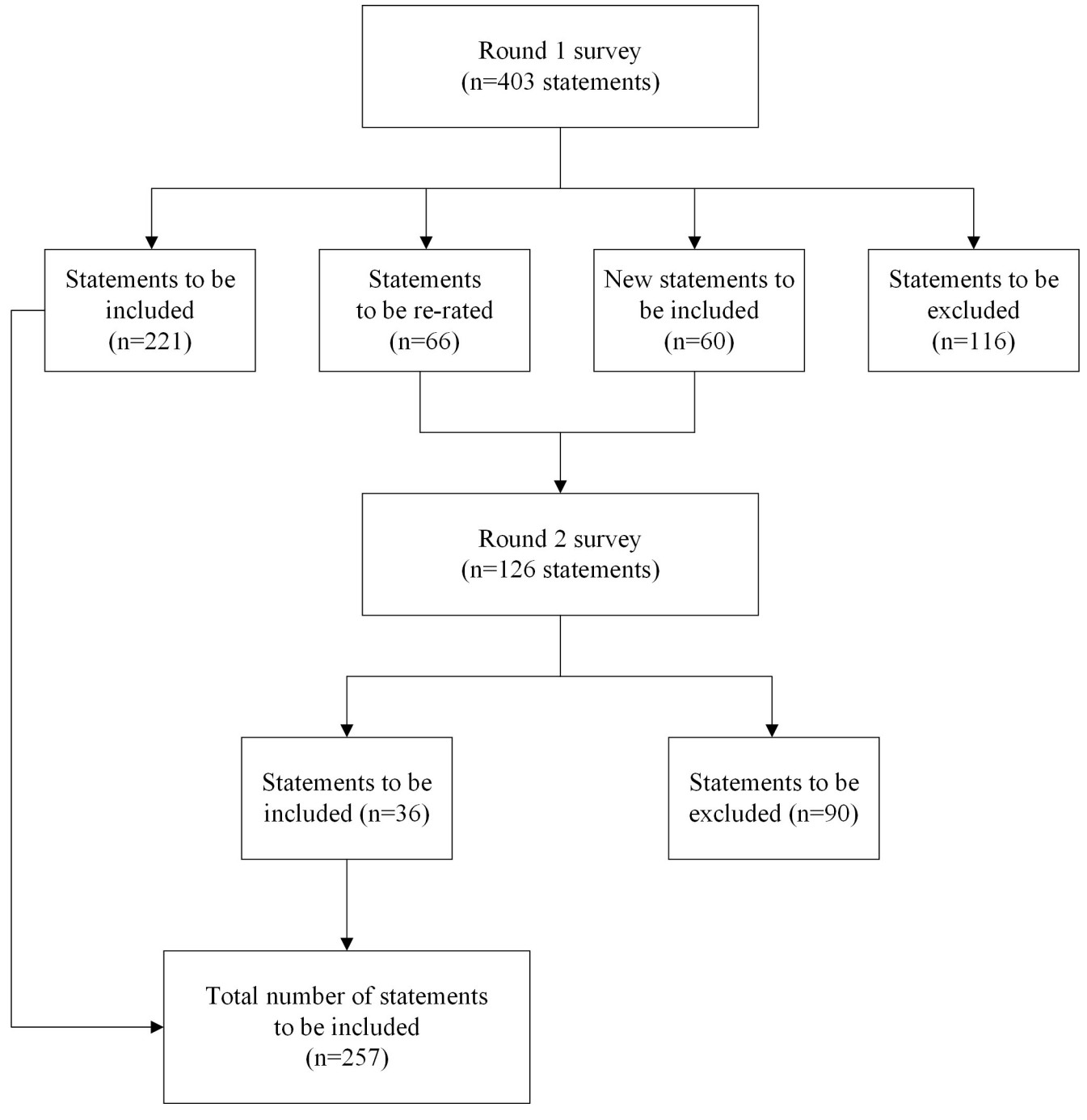

**Fig 1. Overview of the study rounds.**

- **When the person is in crisis (is in a severe psychotic state or behaving aggressively)** (n = 2):

  6. The first aider should avoid using their mobile phone while helping the person.

  7. If police or emergency services personnel arrive, the first aider should be available to explain the person's situation.

- **Aggression (crisis situation)** (n = 1):

    8. If the person is behaving aggressively, the first aider should not leave the person alone.

## Discussion

This study aimed to develop mental health first aid guidelines for psychosis in Brazil using the Delphi expert consensus methodology. The Brazilian guidelines were developed over two survey rounds and involved gathering views and opinions of health professionals and individuals with lived experience of psychosis in Brazil. There were similarities and differences between the Brazilian and English-language guidelines.

### Comparison with the English-language psychosis guidelines

Of the 403 statements from the Round 1 survey, 307 statements (76%) received the same final endorsement ratings (this includes endorsed and excluded statements) in the Brazilian and English-language studies. Of those 307 statements, 204 (66%) statements were endorsed in both guidelines, showing a modest level of similarity between guidelines. For instance, both guidelines emphasized the importance of the first aider staying calm, particularly during a severe episode or aggressive state, with statements like "The first aider should try to be calm when approaching the person, regardless of the person's emotional state" and "The first aider should speak calmly" being endorsed by nearly all participants in both studies.

A key difference between the Brazilian and English-language guidelines was the inclusion of 45 new statements in the Brazilian guidelines beyond the eight new endorsed statements created from panelists' comments. Some of those 45 statements referred to the first aider talking to the person's family or friends, with examples such as "If the first aider is unsure whether the person is experiencing psychosis, they should tactfully ask the person's friends or family whether the person has a diagnosis of psychosis or has experienced psychosis" and "The first aider should ask the person's family or friends if they have noticed any concerning changes in their behavior" only endorsed in the Brazilian study. So, the Brazilian guidelines included more statements related to the involvement of family and friends than the English-language guidelines. This may be due to the importance of family support for people with psychosis in Brazil. A 2018 systematic review found that family was an important factor on initial stages of help-seeking for individuals with mental health issues (including psychotic disorders) in Brazil [23].

Fifty one statements endorsed in the English-language guidelines were not endorsed for the Brazilian guidelines. Some of these statements were related to the first aider openly discussing ways of help-seeking with the person, such as "The first aider should try to find out what type of professional help the person believes will help them" and "The first aider should provide the person with a range of options for seeking professional help". This is in line with our findings from the cultural adaptation of the mental health first aid for suicide in Brazil [18], which also suggested stigma to be a possible barrier for openly discussing help-seeking actions for mental health problems in Brazil.

### Differences between the professional and lived experience panels

There were large differences ($>\pm30\%$) in ratings between the professional and lived experience panels. For instance, panels had conflicting opinions on the extent to which external factors may affect someone with psychosis. Statements such as "The first aider should ask the person if there are any current stressors that may be contributing to their symptoms" and "If the person does not want to seek professional help, the first aider should explore the reasons for this, e.g. not realizing they are unwell, worries about stigma, or not knowing where to get help"

were endorsed by most participants in the professional panel, while only half of the individuals in the lived experience panel endorsed those statements. It is possible that this large difference between panels' ratings of those statements may be due to the considerably lower mental health literacy among individuals with lived experience than health professionals in this study, which may in part explain poor help-seeking behaviors for mental health issues seen in LMICs [7,8].

### Considerations for future use of the Brazilian guidelines

The Brazilian guidelines for psychosis can be used as a stand-alone product as well as a guide for future development of the MHFA training in Brazil. However, it is worth mentioning that although statements can be used individually, these guidelines may be more effective when used to inform a holistic approach to providing mental health first aid.

### Strengths and limitations

The use of the Delphi methodology to develop the Brazilian mental health first aid guidelines for psychosis was a strength in this study. This methodology allowed us to incorporate culturally specific features, such as involvement of family and friends, into these guidelines. Another major strength was the diversity of participants, which in this study included individuals with lived experience in psychosis, either their own or as caretakers, as well as a wide range of ages (19–60 years old). This diversity is a crucial aspect of Delphi studies [19]. However, the study had limitations. One was the low retention rate from Round 1 to Round 2, with less than half of the participants completing the Round 2 survey. Another limitation was the large proportion of women participants, which limited the diversity of the sample. Nevertheless, the drop-out rates were similar between panels, making the ratings comparable.

### Conclusions

A Delphi expert consensus study including health professionals and individuals with lived experience in psychosis was used to develop the Brazilian mental health first aid guidelines for psychosis. Although a modest level of similarity between the Brazilian and English-language guidelines was found, statements related to the importance of family support for people with psychosis and stigma as a potential barrier for discussing help-seeking behaviors for mental health problems were emphasized in the Brazilian guidelines. The Brazilian guidelines also included new statements created and endorsed by local experts. Further research is needed to evaluate the effectiveness of the guidelines in reducing stigma and improving mental health literacy and helping-seeking behaviors in Brazil as well as in the development and implementation of MHFA training in Brazil.

### Supporting information

**S1 File. Survey statements over two survey rounds.** All survey statements in the mental health first aid guidelines for psychosis for Brazil over two survey study rounds (English and Brazilian Portuguese languages).
(XLSX)

**S2 File. Mental health first aid guidelines for psychosis for Brazil.** Final guidelines document in Brazilian-Portuguese language.
(PDF)

## Acknowledgments

We wish to acknowledge the time and effort of the participants, without whom this study would not have been possible.

## Author Contributions

**Conceptualization:** Alexandre Andrade Loch, Nicola J. Reavley.

**Formal analysis:** Simone Scotti Requena, Alexandre Andrade Loch.

**Funding acquisition:** Nicola J. Reavley.

**Methodology:** Nicola J. Reavley.

**Project administration:** Alexandre Andrade Loch, Kathlen Nataly Mendes.

**Supervision:** Alexandre Andrade Loch, Kathlen Nataly Mendes.

**Writing – original draft:** Simone Scotti Requena.

**Writing – review & editing:** Alexandre Andrade Loch, Kathlen Nataly Mendes, Nicola J. Reavley.

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
