## [Decision Letter · Decision Letter 0]

22 May 2024

PONE-D-23-37233Development of mental health first aid guidelines for psychosis for Brazil: a Delphi expert consensus studyPLOS ONE

Dear Dr. Reavley,

Thank you for submitting your manuscript to PLOS ONE. After careful consideration, we feel that it has merit but does not fully meet PLOS ONE’s publication criteria as it currently stands. Therefore, we invite you to submit a revised version of the manuscript that addresses the points raised during the review process.

We look forward to receiving your revised manuscript.

Kind regards,

Vincenzo De Luca

Academic Editor

PLOS ONE

Journal Requirements:

Mental Health First Aid International

Reviewers' comments:

Reviewer's Responses to Questions

**Comments to the Author**

1. Is the manuscript technically sound, and do the data support the conclusions?

Reviewer #1: Partly

2. Has the statistical analysis been performed appropriately and rigorously? 

Reviewer #1: Yes

3. Have the authors made all data underlying the findings in their manuscript fully available?

Reviewer #1: Yes

4. Is the manuscript presented in an intelligible fashion and written in standard English?

Reviewer #1: Yes

5. Review Comments to the Author

**Reviewer #1:** Overall:

This was very clear and interesting and an important topic to research. I commend the author on taking the time to address the implications of cultural differences in assessments/manuals used.

Minor Revisions in Writing style:

Consider revising the first sentence of abstract "Psychosis is highly debilitating." It is very short, not cited and can be incorporated into the next sentence.

Paragraphs in the introduction are somewhat short and can seem choppy.

Some small grammatical errors. The term 'carer' is incorrect (consider caretaker).

Line 84: should not begin new paragraph with "for instance" (can be part of previous paragraph).

Moderately Important Revisions:

Consider the use of the term 'gender' rather than 'sex' and man/woman rather than male/female

In Table 2: Carer (or caretaker) should be moved below 'Age'

Major Revisions:

Given the age range of participants, my concern is that there is a large variance of experience that could have been addressed in the paper. Were there differences in opinion based on level of exposure (i.e. the age ranges are from 19-60)

 Statistical analysis/design:

Line 159: The term 'consensus' should be used (over 80%) to clarify the later use.

More information should be included regarding how participants were selected. There is a section devoted to 'Identification and recruitment of expert panels" (from line 133) but there is not a lot of information. What qualifies them as experts? Does the amount of experience not matter? What measures were taken to ensure that the participants involved were appropriate? How was lived experience with psychosis evaluated (enough to have 1 episode, multiple, etc). Might it be problematic that mostly women participated?

If you recruited participants through word of mouth, might there be some level of similarity in lived experience, shared perspectives, etc.

I am somewhat still unclear as to why the Delphi method was used since it is complex and difficult to keep participants for multiple sessions. Could there be an easier method to get similar (or even more robust) results?

It is mentioned that the Mental health first aid international manual was created with a Delphi technique, but who were the experts for that manual? Are they similar to your experts?

6. PLOS authors have the option to publish the peer review history of their article (what does this mean?). If published, this will include your full peer review and any attached files.

Reviewer #1: No

---

## [Author Response · Author response to Decision Letter 0]

13 Jun 2024

Response to reviewer’s comments:

We are grateful to the editor and reviewer for their comments and the opportunity to revise our submission. We provide our responses below and trust that we have adequately addressed the reviewer’s comments.

We have ensured that our manuscript meets PLOS ONE’s requirements, including adding the role of funder statement in our cover letter, as requested. Additionally, we have made some edits throughout the paper to improve its readability.

Review #1

1. This was very clear and interesting and an important topic to research. I commend the author on taking the time to address the implications of cultural differences in assessments/manuals used.

We thank the reviewer for their comment. 

2. Consider revising the first sentence of abstract "Psychosis is highly debilitating." It is very short, not cited and can be incorporated into the next sentence.

Thank you for this feedback. We agree that the first sentence of the “Abstract” could be improved. We have deleted the statistics and edited the first two sentences of the “Abstract” as follows:

“Psychotic symptoms can be highly debilitating for those experiencing them. Community members, including family and friends, can play a crucial role in providing support to a person during the early stages of psychosis, provided they have the necessary resources.”

3. Paragraphs in the introduction are somewhat short and can seem choppy.

We have made several changes to the “Introduction” to improve the flow. The first four paragraphs (pages 3 and 4) that have been edited are as follows:

“In 2019, the global incidence rate of individuals experiencing psychosis was estimated to be 26.6 per 100,000 person-years, with significant geographical variation [1]. In São Paulo, the largest Brazilian city, the estimated incidence of first-episode psychosis was at 15.8 per 100,000 person-years in 2007 [2], whereas in other municipalities, this was 21.3 per 100,000 person-years more recently [3]. People with psychosis tend to have poorer physical health, feel more isolated and disconnected, have lower rates of employment [5], and face an increased risk of suicide [4], all of which can be highly debilitating for both those affected and those close to them. During psychosis acute states, it carries the highest disability weight among all mental health problems [4]. Therefore, timely support is of extreme importance. 

It is estimated that less than one-third of the global population living with psychosis receive mental health treatment, with far lower coverage in low-and-middle-income countries (LMICs) than in high-income countries (HICs) [6]. One of the major factors contributing to low mental health treatment rates in LMICs is the lack of services or resources. However, poor levels of mental health literacy and high levels of stigma around mental health problems also play crucial roles in the low rates of mental health treatment in LMICs [7-10]. 

Various studies conducted in Brazil have explored stigma and discrimination towards individuals with schizophrenia, commonly associated with psychosis. One study involving 500 participants found that half of them perceived individuals with schizophrenia as capable of provoking negative emotions, while nearly two-thirds viewed them as dangerous [11]. Another study, which included nearly 2,500 participants from the community and psychiatrists, revealed that stigma towards people with schizophrenia increased with the ability to recognize the condition among community members, whereas psychiatrists showed the highest scores in negative stereotyping and perceived prejudice towards schizophrenia [13]. Another study investigated potential risk factors associated with hospital readmission in individuals with bipolar and psychotic disorders [12]. That study found that family’s agreement with continuing hospitalization predicted hospital readmission, with families of readmitted individuals perceiving them as dangerous and unhealthy. These studies provide evidence of the discrimination individuals with psychosis experience in Brazil. Therefore, improving mental health literacy may be a crucial step forward.”

4. Some small grammatical errors. The term 'carer' is incorrect (consider caretaker).

While ‘carer’ is in common use in Australia, where some of the authors work, we have replaced “carer” with “caretaker” throughout the manuscript. 

5. Line 84: should not begin new paragraph with "for instance" (can be part of previous paragraph).

Thanks. We have merged these paragraphs.

6. Consider the use of the term 'gender' rather than 'sex' and man/woman rather than male/female

We have replaced sex (male/female) with gender (men/women) in Table 2 and in text in the “Results” section (paragraph 2, page 9).

“Participant socio-demographic characteristics by panel are shown in Table 2. In the professional panel, most of the participants were women (75%) aged 19–62 (mean ± SD = 36.7 ± 11.5).”

7. In Table 2: Carer (or caretaker) should be moved below 'Age'

We have moved “caretaker” under “age” row in Table 2.

8. Given the age range of participants, my concern is that there is a large variance of experience that could have been addressed in the paper. Were there differences in opinion based on level of exposure (i.e. the age ranges are from 19-60)

Thank you for your comment. We do not think the age range is a limitation, but rather a strength of our study as it only makes the sample more diverse, which is an important consideration in Delphi studies (see Jorm, 2015). Assessing the variance in demographic variables between panels is currently out of scope for our study; however, we agree that the preponderance of women is a limitation of our study. 

We have clarified these points under the following sections:

“Material and methods” section (paragraph 1, page 5):

“This study used the Delphi expert consensus method to develop the Brazilian mental health first aid guidelines for psychosis. This method involves gathering expert opinions and assessing the consensus among groups of individuals with diverse experience in a given field [19]. Although a variety of expertise is not mandatory for the Delphi method, it is crucial to consider diversity when selecting panel members to ensure optimal decision-making, including individuals with lived experience and a diverse range of health professionals [19].”

“Strengths and limitations” section (paragraph 1, page 16):

“The use of the Delphi methodology to develop the Brazilian mental health first aid guidelines for psychosis was a strength in this study. This methodology allowed us to incorporate culturally specific features, such as involvement of family and friends, into these guidelines. Another major strength was the diversity of participants, which in this study included individuals with lived experience in psychosis, either their own or as caretakers, as well as a wide range of ages (19-60 years old). This diversity is a crucial aspect of Delphi studies [19]. However, the study had limitations. One was the low retention rate from Round 1 to Round 2, with less than half of the participants completing the Round 2 survey. Another limitation was the large proportion of women participants, which limited the diversity of the sample. Nevertheless, the dropout rates were similar between panels, making the ratings comparable.” 

9. Line 159: The term 'consensus' should be used (over 80%) to clarify the later use.

We have clarified this point under “Data collection and analysis” (paragraph 2, page 8): 

“All statements were rated on a five-point Likert scale: (1) Essential, (2) Important, (3) Depends/Don’t know, (4) Unimportant, (5) Should not be included. Statements rated as “Essential” or “Important” by 80% or above of participants in each panel were directly included in the final guidelines, as they reached 80% consensus between panels.”

As well as under “Guidelines development” (paragraph 1, page 8):

“Statements that reached consensus at rounds 1 and 2, i.e., those rated as “Essential” or “Important” by >80% participants in both panels, were assembled into sections (see S1 File).”

10. More information should be included regarding how participants were selected. There is a section devoted to 'Identification and recruitment of expert panels" (from line 133) but there is not a lot of information. What qualifies them as experts? Does the amount of experience not matter? What measures were taken to ensure that the participants involved were appropriate? How was lived experience with psychosis evaluated (enough to have 1 episode, multiple, etc). Might it be problematic that mostly women participated?

If you recruited participants through word of mouth, might there be some level of similarity in lived experience, shared perspectives, etc.

We did not have an exclusion criterion for the level of experience in the study. We believe diversity is key in Delphi studies and have clarified this previously. We have clarified this point under “Identification and recruitment of expert panels” (paragraph 2, page 7):

“There were no exclusion criteria regarding the level of experience within panels, to ensure an appropriate level of diversity within each panel. Potential participants were identified and invited by phone, email, or face-to-face to take part in the study. For the professional panel, individuals from universities, including clinicians and researchers, and hospitals specialized in treating people experiencing psychosis (i.e., psychiatry departments of USP and UNIFESP; public universities/hospitals in Brazil) were contacted. For the lived experience panel, outpatients from specialized clinics from USP and UNICAMP, social media support groups, and specialized non-governmental organizations were contacted. To increase the number of participants, confirmed and potential participants were asked to inform others about the study. Potential participants were given a brief description of the study, either verbally or by email, and then sent the survey link.”

11. I am somewhat still unclear as to why the Delphi method was used since it is complex and difficult to keep participants for multiple sessions. Could there be an easier method to get similar (or even more robust) results?

We used the Delphi expert consensus method as this is a valuable method for incorporating the view of diverse experts, while ensuring that all participants have an equal voice. Moreover, we have successfully used this for other mental health problems in Australia, Brazil, China and Chile and Argentina.

12. It is mentioned that the Mental health first aid international manual was created with a Delphi technique, but who were the experts for that manual? Are they similar to your experts?

The MHFA manual is based on guidelines developed with Delphi studies. In each study, Delphi participants were health professionals and people with lived experience.

We have added some text to clarify this point under “Identification and Recruitment of Expert Panels” section (paragraph 1, page 7):

“To ensure a diverse sample, this study had two expert panels: (1) Professional; (2) Lived experience, following the procedures used in Bond et al.’s study conducted in English-speaking countries [20].”

---

## [Decision Letter · Decision Letter 1]

1 Jul 2024

Development of mental health first aid guidelines for psychosis for Brazil: a Delphi expert consensus study

PONE-D-23-37233R1

Dear Dr. Reavley,

We’re pleased to inform you that your manuscript has been judged scientifically suitable for publication and will be formally accepted for publication once it meets all outstanding technical requirements.

Kind regards,

Vincenzo De Luca

Academic Editor

PLOS ONE

Additional Editor Comments (optional):

Reviewers' comments:

Reviewer's Responses to Questions

**Comments to the Author**

1. If the authors have adequately addressed your comments raised in a previous round of review and you feel that this manuscript is now acceptable for publication, you may indicate that here to bypass the “Comments to the Author” section, enter your conflict of interest statement in the “Confidential to Editor” section, and submit your "Accept" recommendation.

Reviewer #1: All comments have been addressed

2. Is the manuscript technically sound, and do the data support the conclusions?

Reviewer #1: Yes

3. Has the statistical analysis been performed appropriately and rigorously? 

Reviewer #1: Yes

4. Have the authors made all data underlying the findings in their manuscript fully available?

Reviewer #1: Yes

5. Is the manuscript presented in an intelligible fashion and written in standard English?

Reviewer #1: Yes

6. Review Comments to the Author

Reviewer #1: I am satisfied with the edits submitted. I believe the authors have met all the requirements and this can move forward to publication.

7. PLOS authors have the option to publish the peer review history of their article (what does this mean?). If published, this will include your full peer review and any attached files.

Reviewer #1: No

---

## [Editor Report · Acceptance letter]

12 Jul 2024

PONE-D-23-37233R1 

PLOS ONE

Dear Dr. Reavley, 

I'm pleased to inform you that your manuscript has been deemed suitable for publication in PLOS ONE. Congratulations! Your manuscript is now being handed over to our production team.

Kind regards, 

on behalf of

Dr. Vincenzo De Luca 

Academic Editor

PLOS ONE